# Health-Related, Social and Cognitive Factors Explaining Gambling Addiction

**DOI:** 10.3390/healthcare11192657

**Published:** 2023-09-30

**Authors:** Javier Esparza-Reig, Manuel Martí-Vilar, Francisco González-Sala, César Merino-Soto, Gregorio Hernández-Salinas, Filiberto Toledano-Toledano

**Affiliations:** 1Department of Psychology, Universidad Europea de Valencia, Passeig de l’Albereda, 7, 46010 Valencia, Spain; javier.esparza@universidadeuropea.es; 2Department of Basic Psychology, Faculty of Psychology, Universitat de València, 46010 Valencia, Spain; manuel.marti-vilar@uv.es; 3Departamento de Psicología Evolutiva y de la Educación, Facultad de Psicología y Logopedia, Universitat de València, 46010 Valencia, Spain; francisco.gonzalez-sala@uv.es; 4Instituto de Investigación de Psicología, Universidad de San Martín de Porres, Av. Tomás Marsano 232, Lima 34, Peru; sikayax@yahoo.com.ar; 5Zongolica-Extensión Tezonapa, Tecnológico Nacional de México, Km. 4 Carr. a La Compañia S/N, Tepetitlanapa, Veracruz 95005, Mexico; gregorio_18_18@live.com.mx; 6Unidad de Investigación en Medicina Basada en Evidencias, Hospital Infantil de México Federico Gómez, Dr. Márquez 162, Doctores, Cuauhtémoc, Mexico City 06720, Mexico; 7Unidad de Investigación Multidisciplinaria en Salud, Instituto Nacional de Rehabilitación Luis Guillermo Ibarra Ibarra, México-Xochimilco 289, Arenal de Guadalupe, Tlalpan, Mexico City 14389, Mexico; 8Dirección de Investigación y Diseminación del Conocimiento, Instituto Nacional de Ciencias e Innovación para la Formación de Comunidad Científica, INDEHUS, Periférico Sur 4860, Arenal de Guadalupe, Tlalpan, Mexico City 14389, Mexico

**Keywords:** gambling, depression, coping, priming, prosocial behavior, cognitive distortion

## Abstract

Background: Gambling addiction was the first addictive behavior not related to substance use that was recognized by the DSM-5. It shares diagnostics and comorbidity with other addictions. Extensive studies have investigated the clinical variables involved, but there have been fewer studies of related cognitive and social variables. In this research, an integrative model was developed to advance the understanding of gambling addiction, and an explanatory model was created based on the concept of cognitive distortions. Methods: The sample comprised 258 university students (59.5% women) with a mean age of 20.95 years (SD = 2.19). A series of questionnaires were administered to measure gambling addiction, depression, coping with stress, prosocial behavior, susceptibility to priming and cognitive distortions about gambling. In addition, correlations, multiple linear regressions and a simple mediation model of these variables were analyzed. Results: The results indicated that gambling addiction was correlated with a variety of clinical, social and cognitive factors. These factors contributed to a model that predicted 16.8% of the variance in gambling addiction and another model using cognitive distortions as a predictor and the maximum bet as a mediator that predicted 34.5% of the variance. Conclusions: The study represents an advance by developing a theoretical model from an integrative perspective and providing a new explanatory model. The findings of this research are of great importance in the development of prevention and intervention programs for gambling addiction.

## 1. Introduction

Gambling addiction is a maladaptive and persistent gambling pattern that generates lasting clinical problems [1]. Proof of the importance of this pathology is that it was the first addictive behavior that does not involve substance use that was recognized in the DSM-5 [2].

In the DSM-5 [3], gambling disorder is included as a substance-related and addictive disorder; it is characterized by gambling behavior that persists over time, is maladaptive and generates clinically significant distress that is not explained by the presence of a manic episode. As stated in criterion A, a person with pathological gambling shows at least four of the following criteria in the last 12 months: 1. He or she feels the need to gamble increasing amounts of money to achieve the desired excitement. 2. He or she is nervous or irritated when trying to reduce or quit gambling. 3. He or she has made repeated efforts to control, reduce or abandon gambling, always without success. 4. Often, his or her mind is occupied with gambling. 5. He or she often gambles when he or she feels uneasy. 6. After losing money in gambling, he or she usually comes back another day to try to win. 7. He or she lies to hide his or her degree of involvement in gambling. 8. He or she jeopardized or lost an important relationship, a job or an academic or professional career because of gambling. 9. He or she relies on others to give him or her money to alleviate his or her desperate financial situation caused by gambling.

According to the indicators included in criterion A, gambling is considered problematic when the person meets between one and three indicators. With four or more criteria met, gambling is considered to be disordered; the disorder is considered mild when the individual meets between four and five indicators, moderate when he or she meets between six and seven indicators and severe when he or she meets all indicators.

Problem gambling and gambling disorder present a similar diagnosis and high comorbidity with substance addictions. Specifically, pathological gambling shares some characteristics with substance addictions, such as craving, loss of control, withdrawal syndrome and tolerance, understood as the need to play or consume more [4], in addition to not being related to an obsessive-compulsive disorder [5,6].

Additionally, compared to the general population, people with a possible gambling disorder or problem gambling tend to present a higher expenditure of money wagered in a single day [7]. Responsible gaming programs are used in many places to prevent, or at least reduce, gambling problems [8]. Research in this area reinforces the importance of establishing public health policies for the prevention and reduction of gambling problems [9]. Choliz and Saiz-Ruiz [10] carried out a review of these issues and designed a proposal for government regulation of gambling in Spanish society.

### 1.1. Current State of Gambling Addiction

Calado and Griffiths [11], studying this problem, carried out a systematic review of the literature to analyze the prevalence of problem gambling. This review took articles published since 2000 from any country that published prevalence rates. They found that between 0.1 and 5.8% of participants had presented problem gambling in the last year and that between 0.7 and 6.5% had presented so throughout their lives. Analyzing the prevalence by continents, they found that in North America, the figure ranged between 2 to 5%; in Asia, it ranged between 0.5 and 5.8%; in Oceania, it ranged between 0.4 and 0.7%; and in Europe, it ranged between 0.1 and 3.4%. Calado et al. [12] conducted a similar review focusing on the prevalence of problem gambling in adolescents since 2000. In the studies in that review, between 0.2 and 12.3% of participants met the criteria for problem gambling of the Diagnostic and Statistical Manual-IV adapted format for Juveniles (DSM-IV-J) [13] and other instruments; however, this study did not differentiate between problem gambling and pathological gambling when analyzing the prevalence.

In the Spanish context, according to data from the survey on Alcohol and Drugs in Spain, EDADES 2022 [14], 1.7% of those surveyed between 15 and 64 years of age presented problem gambling or gambling disorder according to the DSM-5 criteria. This percentage was higher for men (2.4%) than women (0.9%). Considering the number of DSM-5 indicators, 2.2% presented problem gambling by meeting between one and three indicators. On the other hand, 0.4% presented a possible gambling disorder by meeting four or more DSM-5 indicators. Extrapolating these data to the Spanish population between 15 and 64 years of age, 1.3% could present problem gambling, and 0.4% could have a possible gambling disorder.

In the case of adolescents between 14 and 18 years of age, according to a 2021 survey, 17.9% could have a gambling disorder based on the answers given in the Lie/Bet scale by Johnson et al. [15]. Extrapolating these data to the entire population between 14 and 18 years of age, 3.4% could have a possible gambling disorder, and this percentage would be higher for men (5.0%) than women (1.9%).

### 1.2. Relationship of Gambling Addiction to Clinical and Health Factors

Given the relevance of gambling problems, psychologists have studied their relationship with various constructs, especially with psychopathologies and other health-related factors. Depression is the mental health pathology with the highest prevalence rate worldwide [16] and shows one of the strongest connections with gambling addiction. These two pathologies have high comorbidity, which makes the study of their relationship highly relevant for understanding problem gambling [17]. Depression directly correlates with gambling behavior, and, deepening this relationship, it has been found that people with gambling addiction problems have higher rates of depression and that depression acts as a predictor of gambling addiction [17,18].

In relation to stress, there are differences in the ways in which people cope with stress depending on whether they have problems with gambling [19]. Gambling problems appear to be associated both with higher levels of maladaptive techniques of coping with stress [20] and with lower levels of adaptive coping techniques [21]. In the clinical setting, people with gambling addiction also seem to cope with stressful situations using maladaptive techniques, such as trying to avoid stress-generating situations instead of facing them to find solutions [22,23]. The cognitive distortions that people have toward gambling and their maladaptive strategies for coping with stress seem to mediate the relationship between depression and gambling addiction. Therefore, when both pathologies occur comorbidly, one possible avenue for interventions would be to focus on these cognitive distortions [24,25].

### 1.3. Relation of Gambling Addiction with Social and Cognitive Factors

Contrary to the research interest in clinical factors, there is a lack of exhaustive research on the relationship of gambling addiction with certain social factors, such as prosocial behavior, and cognitive factors, such as the priming effect. Prosocial behavior consists of actions that are aimed at promoting cooperation, tolerance, help and solidarity and that are related to the prevention of behavioral problems, such as antisocial and criminal behavior [26,27]. Pathological gamblers seem to exhibit this type of behavior to a lesser extent than the general population, and prosocial behavior, in addition to other emotional, behavioral and social factors, is also a predictor of problems with gambling [28].

In the study of human behavior, prospect theory proposed by Tversky and Kahneman [29] identifies a phenomenon known as the framing effect; the theory suggests that the different ways in which a problem is framed can influence the decisions that people make about that problem. According to this theory, people are guided by risk aversion and prefer a safe alternative to a riskier alternative when the alternatives are defined as potential gains rather than when a context-dependent benchmark is used. For Giuliani et al. [30] and Manippa et al. [31], people value a gain in the form of money, accounting for aspects of an affective or utilitarian nature. On this basis, in a win-win situation, such as winning money in gambling, the individual will make a decision thinking about the benefits or utility of the win, which will lead him or her to choose the least risky option. On the other hand, when the alternatives are focused on potential losses, the person will take more risks in their decisions.

In this sense, the decisions made by the individual are influenced by the context in which they are presented. Thus, if we want to encourage gambling, in accordance with the framing effect, the individual should be exposed to a greater extent to messages related to what he or she can gain from gambling and to a lesser extent to messages related to what he or she can lose.

Different from the framing effect is the priming effect, which accounts for previous experiences or stimuli to which the person has been exposed when responding to a given situation. Cesario [32] defined the priming effect as an implicit memory effect. This effect acts as a conditioning. In the specific case of gambling, it is to be expected that positive experiences associated with gambling, such as winnings or benefits at economic, social, psychological levels, will generate a greater predisposition to continue gambling. In the case of alcohol consumption, different studies have pointed out how initial consumption can motivate subsequent consumption, which would occur both in drinkers and in those who try to abstain [33,34].

Applying this framing effect to economic issues, it has been found that people tend to express more aversion to loss than the profits they can make; that is, people value what they already have more than what they can obtain, so under different framings, they tend to choose the option that minimizes losses [35,36]. Takeuchi et al. [37] found that a group of problem gamblers and a control group of healthy people showed no differences in their loss aversion.

### 1.4. The Role of Cognitive Distortions in Gambling

One of the most noteworthy traits of pathological gamblers is their distorted cognition in relation to gambling [38,39]. This relationship is not clearly defined, with some theories arguing that the severity of problematic gambling and the motivations that lead people to gamble act as predictors of cognitive distortions about gambling [38], while other theories posit that cognitive distortions are predictors of future gambling behavior [40]. Depending on which of the two theories is favored, it can be determined whether it is more effective to focus interventions on gambling behavior or on players’ cognition about gambling [40].

### 1.5. The Present Study

The objectives of this study are, first, to outline the differences among clinical, social and cognitive factors to develop an integrative model that helps better understand the problem of gambling addiction. Second, an attempt is made to create an explanatory model of gambling addiction using, in this case, closely related factors, such as cognitive distortions about gambling and the maximum amount of money wagered.

Based on those objectives, we propose the following hypotheses:

**Hypothesis 1.** *There will be a relationship between gambling addiction and clinical, social and cognitive factors. Specifically, the following secondary hypotheses are proposed*:

**Hypothesis 1.1.** *Gambling addiction will be positively related to depression, as stated by* [17,18].

**Hypothesis 1.2.** *Gambling addiction will be positively related to more maladaptive strategies for coping with stress, as reported by* [20,21,22,23].

**Hypothesis 1.3.** *Gambling addiction will be positively related to greater susceptibility to priming, greater cognitive distortions [38,39] and, as suggested by [7,38], more money wagered*.

**Hypothesis 1.4.** *Gambling addiction will be negatively related to prosocial behavior, as suggested by* [28].

**Hypothesis 2.** *Clinical, social and cognitive factors will all be predictors of gambling addiction*.

**Hypothesis 3.** *Cognitive distortions about gambling will be predictors of gambling addiction, with the maximum amount wagered playing a mediating role*.

## 2. Materials and Methods

### 2.1. Participants

A total of 258 university students (153, or 59.5%, were women) between the ages of 18 and 26 completed the study questionnaires, and the mean age of the total sample was 20.95 years (SD = 2.19). Table 1 shows the demographic characteristics of the sample.

### 2.2. Measurement Instruments

To measure gambling addiction, the version of the South Oaks Gambling Screen (SOGS [41]) that was validated in the Spanish population [42] was applied. This instrument consists of 20 items (mostly dichotomous); the 3 initial items are not considered in the total score and are used to assess the type of gambling or betting, the maximum amount wagered and whether any close contacts of the participant have problems with gambling. The score of the validated Spanish version ranges from 0 to 19, with the authors considering a score greater than 4 as indicative of problems with gambling. In the original version, all the items evaluate addiction to gambling throughout the lifespan of the participant. The Cronbach’s alpha value for this questionnaire was 0.8. Additionally, the participants indicated the maximum amount of money they had wagered at one time.

The Basic Depression Questionnaire (CBD [43]), also validated in the Spanish population, was used to measure depression. This instrument measures a series of 21 symptoms typical of depression. Subjects must respond indicating when they have experienced the symptoms, with the options being “never”, “for weeks”, “for months” and “for years”. A Likert-type response scale is used, with higher scores associated with higher depression rates. Recently, this questionnaire has shown good psychometric properties and validity for the specific diagnosis of depressive disorders versus anxiety disorders [16]. The Cronbach’s alpha value for this questionnaire was 0.87.

Different styles of coping with stress were measured using the Stress Coping Questionnaire (SCQ [44]), which has been validated in the Spanish population. This questionnaire comprises 42 items grouped into 7 subscales that measure different ways of coping with stress: focusing on solving the problem, negative self-targeting, positive reevaluation, open emotional expression, avoidance, seeking social support and religion. The response scale was a Likert-type scale with five alternatives, with higher scores indicating a greater tendency to use a certain type of coping. The Cronbach’s alpha value for each of the subscales were as follows: 0.83 for focusing on solving the problem, 0.68 for negative self-targeting, 0.73 for positive reevaluation, 0.71 for open emotional expression, 0.69 for avoidance, 0.91 for seeking social support and 0.95 for religion.

To measure prosocial behavior, the Prosociality Scale by Caprara et al. [45] was used. This scale serves to measure prosocial behavior in youth and adults, differentiating more prosocial from less prosocial individuals, based on the scale of prosocial behavior for children by Caprara and Pastorelli [46]. In the current study, an adapted version used by Martí-Vilar et al. [47] was deployed. This scale is made up of 16 Likert-type items with 5 response alternatives ranging from 1 (“never/almost never”) to 5 (“always/almost always”). The Cronbach’s alpha value for this scale was 0.89.

Parts 2 and 4 of the inventory developed by Lepore [48] were applied to assess the susceptibility of the participants to priming effects. A series of five scenarios is presented (for each of the two parts) in which participants have to make decisions about economic issues or life and death issues. The two parts parallel each other, with each presenting the same situations but changing the frame. The answers are dichotomous, and if a participant selects a different option in each of the parallel scenarios, it implies that he or she is making inconsistent decisions due to the priming effect. Therefore, the closer the scores are to 1, the greater the individual’s susceptibility to priming.

Finally, to assess the participants’ cognitive distortions about gambling, the Spanish version of the Gambling-Related Cognition Scale (GRCS [49]) was used. This scale is made up of 23 Likert-type items with 7 alternatives ranging from 1 (“completely disagree”) to 7 (“completely agree”). It is structured in five subscales that assess cognitive biases associated with gambling (gambling expectations, illusion of control, prediction of control, inability to stop playing and interpretive bias). The total score across all subscales was used in the current research. For both the subscale scores and the total score, higher scores indicate greater cognitive distortions. The Cronbach’s alpha value for the total scale was 0.92.

Each study variable was measured by a single measure. The most conceptually close, but still different, symptoms were the measures of general symptoms and cognitive distortions.

### 2.3. Procedure

This cross-sectional study is part of a larger study that seeks to explain the functioning of gambling addiction and its consequences. The Commission for Ethics in Experimental Research of the University of Valencia approved this study (procedure number 1040164). Data were collected between the months of May and December 2019 from students in different faculties of the University of Valencia. The participants were contacted through their teachers; they completed the questionnaire on paper and always in the presence of one of the researchers to guarantee an appropriate environment for carrying out the questionnaire and to ensure that any queries from the participants were answered. The questionnaire required approximately 50–60 min to complete. After participating in the research, all participants signed an informed consent form describing the conditions of the research and explaining that the data collected would be completely anonymized. No incentives were offered to the participants.

### 2.4. Ethical Considerations

This study is a part of the research project HIM/2015/017/SSA.1207 “Effects of mindfulness training on psychological distress and quality of life of the family caregiver”, which was approved by the Research, Ethics and Biosafety Commissions of the Hospital Infantil de México Federico Gómez National Institute of Health in Mexico City. While conducting this study, we followed the ethical rules and considerations for research with humans currently enforced in Mexico [50] and those outlined by the American Psychological Association [41]. All family caregivers were informed of the objectives and scope of the research and their rights according to the Helsinki Declaration [51]. The participants who agreed to participate in the study signed an informed consent letter. Participation in this study was voluntary and did not involve payment.

### 2.5. Analysis

First, the distribution and frequency of responses for each of the measured variables were analyzed. Next, Pearson correlation analyses were performed to explore the relationship between the different measured variables and gambling addiction. Subsequently, a series of simple and multiple regression analyses were carried out between the relevant variables and, finally, a simple mediation model was analyzed. Statistical analyses were performed using SPSS 20.0 statistical software, using an additional macro for the mediation analysis.

## 3. Results

### 3.1. Relation of the Various Factors with Gambling Addiction

Table 2 shows the Pearson correlations between each of the factors studied and gambling addiction. The only significant negative correlation was between gambling addiction and prosocial behavior (r = −0.13, *p* < 0.05). The strongest significant positive correlations occurred between gambling addiction and the maximum amount of money wagered (r = 0.52, *p* < 0.01) and cognitive distortions about gambling (r = 0.50, *p* < 0.01). For the other variables that showed significant correlations with gambling addiction, the strongest was susceptibility to priming (r = 0.25, *p* < 0.01), followed by depression (r = 0.20, *p* < 0.01), coping with stress through religion (r = 0.18, *p* < 0.01) and coping with stress through open emotional expression (r = 0.14, *p* < 0.05).

### 3.2. Regressions on Gambling Addiction

First, a multiple linear regression was performed in which the dependent variable (DV) was gambling addiction, and the independent variables (IVs) were the health, social and cognitive factors that had shown significant correlations with the DV. Using the successive step or stepwise method to perform multiple linear regression, coping with stress through open emotional expression was eliminated first, as it was not significant in the model. Therefore, the final model was made up of gambling addiction as the DV and depression, religion as a way of coping with stress, prosocial behavior and priming susceptibility as the IVs.

The regression model was statistically significant, F(4) = 8.08, *p* < 0.01, and all the IVs in the model were also significant predictors (see Table 3). The corrected R2 value was 16.8%, indicating the proportion of the variance in gambling addiction explained by the variance in the predictor variables. Analysis of the residuals indicated that the data fit well with the assumptions of the linear regression model.

Furthermore, a simple linear regression was estimated in which gambling addiction was the DV and cognitive distortions were the IV. The slope of the regression was statistically significant, β = 0.5, t(1) = 9.32, *p* < 0.01, so it was accepted that there was a linear relationship between gambling addiction and cognitive distortions. The corrected R2 value was 25.1%, indicating that approximately a quarter of the variance in gambling addiction was explained by variance in cognitive distortions. In this case, the data also fit the assumptions of a linear regression model.

Finally, a simple mediation model with bootstrapping with 1000 samples, shown in Figure 1, was tested in which cognitive distortions acted as predictors of gambling addiction, with the maximum amount of money wagered being a mediator in this relationship. The model explained 34.5% of the variance (*p* < 0.01), and the indirect effect was 0.39 (*p* < 0.01). All the relationships shown were statistically significant (*p* < 0.1), and the effect increased considerably when the maximum amount of money bet as a mediator was added.

## 4. Discussion

In this research, we sought to increase knowledge of gambling addiction, specifically in relation to other factors, covering different areas to develop an integrative model that did not leave out any field. Based on this, it was first hypothesized that gambling addiction would have positive relationships with depression, maladaptive stress-coping techniques, susceptibility to priming, cognitive distortions about gambling and the maximum amount of money wagered. Furthermore, the relationship between gambling addiction and prosocial behavior was predicted to be negative (Hypothesis 1).

In the case of depression, the results obtained were consistent with hypothesis 1.1 since the two factors were correlated positively and significantly. This relationship (as well as the relationship between gambling addiction and other pathologies or clinical problems) has been widely addressed, and these results are consistent with other recent investigations that have also found direct relationships between gambling addiction and depression [17,18,24,25].

Additionally, in the results of this research, gambling addiction showed a positive relationship with a maladaptive way of coping with stress (Hypothesis 1.2), namely, resorting to magical, quasi-religious thinking instead of confronting one’s problems [44]. Contrary to our hypothesis, gambling addiction also showed a positive correlation with the adaptive technique of open emotional expression. The remaining coping techniques measured by the SCQ did not show any statistically significant correlations with gambling addiction. The literature on these relationships has reported that there are differences in people’s coping with stress depending on whether they have a problem with gambling [19], which tends to lead those with problem gambling to use higher levels of maladaptive techniques [20,21,22] and lower levels of adaptive techniques [21].

Regarding priming susceptibility (Hypothesis 1.3), the results indicated that higher levels of susceptibility correlated directly with higher levels of gambling addiction. Specifically, the priming situation analyzed was one of those collected in the questionnaire of Lepore [48], extracted from Kahneman et al. [52]. There is not much literature on this topic since existing studies have focused more on loss aversion. Takeuchi et al. [37] found no difference in loss aversion between a group of healthy subjects and another group of pathological players, with the limitation that this result came from a sample of only 57 participants. Gambling addiction was also found to correlate positively with both cognitive distortions about gambling and the maximum amount of money wagered (Hypothesis 1.3), corroborating previous research [7,38]. Finally, in line with what was hypothesized (Hypothesis 1.4), gambling addiction and prosocial behavior correlated in a negative and statistically significant way, as in the literature [43].

In short, the results supported the first hypothesis except for the sub hypothesis related to coping with stress since, contrary to what was expected, gambling addiction correlated positively with an adaptive type of coping and showed a relationship with only one type of maladaptive coping. Second, it was hypothesized that clinical, social and cognitive factors would work together as predictors of gambling addiction (Hypothesis 2). The data obtained corroborated this hypothesis since an integrative model was obtained that combined clinical (depression and stress), cognitive (susceptibility to a priming effect) and social (prosocial behavior) factors, explaining more than 16% of the variance in gambling addiction. Previously, the predictive capacity of depression [17,18], maladaptive techniques for coping with stress [20,21] and, to a lesser extent, prosocial behavior was studied [28], while it had not been found before that susceptibility to priming effects could be a predictor of gambling addiction [37]. The present study, on the one hand, highlights this predictive capacity of priming susceptibility and, on the other hand, offers an integrative approach to help us understand the variables that affect gambling addiction.

Finally, it was hypothesized that cognitive distortions would be predictive of gambling addiction, with this relationship mediated by the maximum amount of money wagered (Hypothesis 3). This hypothesis holds with the data obtained since, while a simple linear regression model with cognitive distortions as the IV explained approximately a quarter of the variance in gambling addiction, the mediation model, including money wagered, increased the proportion of variance explained to just over a third. Existing research on cognitive distortions and gambling addiction thus far has two main currents, with some authors indicating that distortions are predictors of addiction [40] and others arguing that gambling addiction acts as a predictor of cognitive distortions [38]. This study follows research such as that of Yakovenko et al. [40] in proposing an innovative model to contribute to our knowledge about gambling addiction.

### Study Limitations

A limitation of the current study is that it did not consider additional clinical, social and cognitive variables that could be involved in gambling addiction. This could be corrected in future studies. Among these variables, the presence of attention deficit hyperactivity disorder should be accounted for, examining the relationship with addictive behaviors, as indicated by Fatséas et al. [53], as well as with autistic traits [54], since there is a relationship with the prevalence of internet gaming disorder. Another limitation of the study is related to the small sample size and the fact that the sample included only university students, which limits the generalizability of the results. In addition, the lack of a clinical sample should be considered another limitation of the study. Finally, the present study did not account for emotional variables that have been related to addictions, such as emotional codependence in the case of internet addiction [55]. It would also be interesting to systematically review the different gambling addiction prevention and intervention programs to analyze which ones work best and to try to identify the characteristics that make them more effective, relating these to the findings obtained in studies such as this one. In short, with this research, progress has been made in the study of the problem of gambling addiction through novel integrative models, which may be interesting to the field of clinical psychology.

## 5. Conclusions

From the data obtained in this study, it can be concluded that it is necessary to try to contemplate all the variables involved in gambling addiction rather than focusing only on clinical factors to design more comprehensive intervention programs that work simultaneously on all affected dimensions in people who may present problems with gambling. On the other hand, the results support the relevance of working on the cognitive distortions that people have about gambling with the aim of preventing or reducing their gambling addiction problems and working on the behavioral aspect (the maximum amount of money they have wagered), which has been shown to mediate this relationship. These contributions could be interesting both for the development of intervention programs in people who present gambling problems and in the design of prevention plans to work with people at risk.

The results indicated that gambling addiction correlates with a variety of clinical, social and cognitive factors, with these factors contributing to a model that predicted 16.8% of the variance in gambling addiction and another model that predicted 34.5% of the variance using cognitive distortions as a predictor and maximum bet as a mediator. The results represent an advance, first, by developing a theoretical model from an integrative perspective and, second, by providing a new explanatory model. The findings of this research are of great importance in the development of prevention and intervention programs for gambling addiction.

## Figures and Tables

**Figure 1 healthcare-11-02657-f001:**
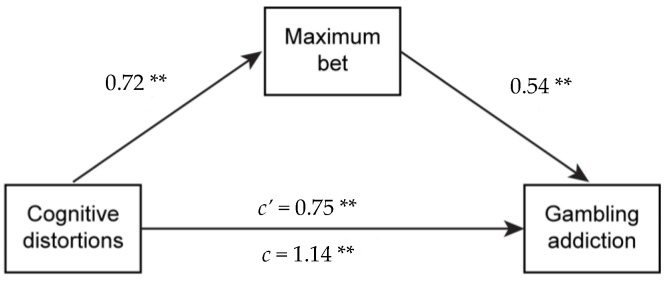
Mediated regression model. ** *p* < 0.1; *c’* = direct effect; *c* = effect with mediator. Bootstrap 1000 samples.

**Table 1 healthcare-11-02657-t001:** Demographic characteristics of the sample.

Characteristics	Total (n = 258)
Women, n (%)	153 (59.5)
Mean age (SD)	20.95 (2.19)
Nationality, n (%)	
Spanish	249 (96.5)
Others	9 (3.5)
Relationship status, n (%)	
Single	150 (58.2)
In a relationship	102 (39.5)
Married	6 (2.3)
Work situation, n (%)	
Unemployed	183 (71)
Part-time	66 (25.5)
Full-time	9 (3.5)

**Table 2 healthcare-11-02657-t002:** Correlations between gambling addiction and the other factors.

Factors	SOGS
Depression	**0.2** **
Focusing on solving the problem	−0.01
Negative self-targeting	0.05
Positive re-evaluation	−0.06
Open emotional expression	**0.14** *
Avoidance	−0.07
Seeking social support	−0.1
Religion	**0.18** **
Prosocial behavior	−**0.13** *
Susceptibility to priming	**0.25** **
Cognitive distortions about gambling	**0.5** **
Maximum sum wagered	**0.52** **

Notes: Significant correlations in bold. * *p* < 0.05. ** *p* < 0.01.

**Table 3 healthcare-11-02657-t003:** Multiple linear regression model.

Predictor	β	T	*p*
Depression	0.24	3.01	<0.01
Priming susceptibility	0.25	3.27	<0.01
Prosocial behavior	−0.18	−2.28	<0.05
Religion	0.17	2.18	<0.05

## Data Availability

The raw data supporting the conclusions of this article will be made available by the authors without undue reservation.

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
