# Peer review of "Health-Related, Social and Cognitive Factors Explaining Gambling Addiction"

_healthcare, 2023, doi:10.3390/healthcare11192657_

Round 1
Reviewer 1 Report
The study shows the correlation between gambling addiction and several clinical, social, and cognitive factors in a group of university students.
The suggestions I've made are intended to enhance the clarity, depth, and comprehensiveness of the paper.
Introduction section.
It might be appropriate to provide official data on the incidence of gambling disorder, and perhaps more specific data on the number of children and adolescents affected This would give context to why is important to better understand its impact on the mental health of the population.
Line 47: The sentence is not clear, please explain it better.
Line 59: The authors should define what problematic gambling is and the definition of gambling disorder following the DSM 5 criteria.
Line 85: The sentence is not clear, please explain it better.
Line 91. The authors mention the prime effect. They should better describe this phenomenon and better define the difference between priming effect and framing effect.
Line 103. To facilitate reading, it would be appropriate not to present the previous studies in such detail, omitting data such as the description of the sample of these studies, the procedure. It would be better to focus on the results so that the reader does not lose the thread and understands the similarities between these previous studies and the study he has just read.
Line 105. The manuscript provides background on the role of cognitive distortions in gambling. I would suggest considering studies about correlation between attention deficit disorders and gambling disorder. Some examples are as follows:
doi: 10.1016/j.psychres.2016.10.071.
doi: 10.3390/brainsci11060774.
Line 126 Please correct the sentence.
As a final suggestion, at the end of the introduction, after stating the objective of the study, it would be appropriate to state the hypotheses of the study. This would help the reader to understand the relationship between the theoretical framework and the analyses performed.
Materials and method section
The study design has been described in detail, describing the procedure followed to obtain the sample. The study population has been adequately described. Did the authors used social media for the recruitment process?
The description of the survey is adequate, explaining in detail the items used to obtain information on each variable, also explaining the psychometric properties of each instrument. It may be necessary to explain why these items were selected from different scales to measure the same variable.
Results section
The results on sociodemographic data are complete and adequately presented. However, the authors should report how many students from the sample showed gambling addiction.
The rest of the results are adequately presented, both in the text and in the tables.
Discussion section
The discussion correctly summarizes the results obtained and compares them with the theoretical background of this field of research.
I recommend English editing
Author Response
Comments and Suggestions for Authors
The study shows the correlation between gambling addiction and several clinical, social, and cognitive factors in a group of university students.
The suggestions I've made are intended to enhance the clarity, depth, and comprehensiveness of the paper.
Introduction section.
It might be appropriate to provide official data on the incidence of gambling disorder, and perhaps more specific data on the number of children and adolescents affected This would give context to why is important to better understand its impact on the mental health of the population.
We thank the reviewer for this observation. As suggested by the reviewer, data on problem gambling and gambling disorder in the Spanish context and in adolescents have been provided in the modified manuscript. This information has been added at the end of section 1.1.
Line 47: The sentence is not clear, please explain it better.
We thank the reviewer for this observation. In the revised manuscript, the sentence has been modified and a new citation has been included.
Line 59: The authors should define what problematic gambling is and the definition of gambling disorder following the DSM 5 criteria.
Thank you very much for this reviewer's comment. In the modified manuscript, the diagnostic criteria for pathological gambling disorder and problem gambling have been included at the beginning of the introduction.
Line 85: The sentence is not clear, please explain it better.
We appreciate the reviewer's comment. In the modified manuscript, the sentence has been reworded more clearly.
Line 91. The authors mention the prime effect. They should better describe this phenomenon and better define the difference between priming effect and framing effect.
We thank the reviewer for this observation. In the modified manuscript, we have specified what is meant by framing effect and priming effect.
Line 103. To facilitate reading, it would be appropriate not to present the previous studies in such detail, omitting data such as the description of the sample of these studies, the procedure. It would be better to focus on the results so that the reader does not lose the thread and understands the similarities between these previous studies and the study he has just read.
We thank the reviewer for this observation. We hope that it will now be easier to read.
Line 105. The manuscript provides background on the role of cognitive distortions in gambling. I would suggest considering studies about correlation between attention deficit disorders and gambling disorder. Some examples are as follows:
doi: 10.1016/j.psychres.2016.10.071.
doi: 10.3390/brainsci11060774.
We thank the reviewer for this observation. In the modified manuscript, these studies have been included in the discussion section, when discussing future studies in which other variables should be included.
Line 126 Please correct the sentence.
As a final suggestion, at the end of the introduction, after stating the objective of the study, it would be appropriate to state the hypotheses of the study. This would help the reader to understand the relationship between the theoretical framework and the analyses performed.
We appreciate the reviewer's comment. In the modified manuscript, as suggested by the reviewer, the number of hypotheses has been specified and related to the theoretical framework. This has also been transferred to the discussion section.
Materials and method section
The study design has been described in detail, describing the procedure followed to obtain the sample. The study population has been adequately described. Did the authors used social media for the recruitment process?
How the participants were contacted through their teachers has been included in the procedure
The description of the survey is adequate, explaining in detail the items used to obtain information on each variable, also explaining the psychometric properties of each instrument. It may be necessary to explain why these items were selected from different scales to measure the same variable.
Different instruments were not used to measure a single variable. The closest thing was the use of the SOGS and the GRCS, but both were applied because one focused on the general symptoms and the other on the cognitive distortions associated with it.
Results section
The results on sociodemographic data are complete and adequately presented. However, the authors should report how many students from the sample showed gambling addiction.
It has not been reported how many people were addicted to gambling because this was not the objective of the investigation. This has focused on problematic gambling behavior, without the need to reach an addiction diagnosis or an X score on the SOGS.
The rest of the results are adequately presented, both in the text and in the tables.
Thank you for this review.
Discussion section
The discussion correctly summarizes the results obtained and compares them with the theoretical background of this field of research.
Thank you for this review.
Comments on the Quality of English Language
I recommend English editing
Reviewer 2 Report
The current study addresses an important issue which is related to gambling addiction. The introduction discusses the problem under investigation. However, the authors used many measures, most of which were not used for the analysis. I think the authors could improve their current version of the paper by considering other variables in the mediation model and other hypotheses. Finally, since the mediation model uses observed variables, please set the analysis using a bootstrapping approach (5.000 resamples). The current results of the mediation model suggest a partial mediation, which should be discussed. The role of the priming effect should also be included in the analysis. Please also check the data quality in terms of univariate and multivariate distribution.
Author Response
Comments and Suggestions for Authors
The current study addresses an important issue which is related to gambling addiction. The introduction discusses the problem under investigation. However, the authors used many measures, most of which were not used for the analysis. I think the authors could improve their current version of the paper by considering other variables in the mediation model and other hypotheses. Finally, since the mediation model uses observed variables, please set the analysis using a bootstrapping approach (5.000 resamples). The current results of the mediation model suggest a partial mediation, which should be discussed. The role of the priming effect should also be included in the analysis. Please also check the data quality in terms of univariate and multivariate distribution.
All the questionnaires applied have been included in the analyzes carried out. Table 2 shows the analysis with all of them.
Reviewer 3 Report
Dear Author,
First of all thank for the opportunity I was given to review this paper. The topic is interesting and while showing itself to be substantially in line with other studies already published on the subject, it allows to propose to the general public some results related to current issues in the related to gambling addiction and comorbidity with other addictions. The authors presented a research developed an integrative model to advance the understanding of gambling addiction, seeking to create an explanatory model based on cognitive distortions.
The results of the study reaffirm that gambling addiction correlates with a variety of clinical, social and cognitive factors, with these contributing to a model that predicted 16.8% of the variance in gambling addiction, and another model that predicted 34.5% of the variance using cognitive distortions as a predictor and maximum bet as mediator.
The results represent an advance by on the one hand developing a theoretical model from an integrative perspective, and, on the other hand, providing a new explanatory model.
Several points of the work, however, need major revisiting and integration in order to be able to raise the overall level of the contribution. I provide below my suggestions related to the different sections of the article:
1. The abstract definitely needs to be rewritten. The articulation of the different sections is confusing and inaccurate. In particular it should be divided into: Introduction, objectives and methods, results and conclusions
2. Increase the keywords to at least seven.
3. The background introduction is insufficiently developed, lacking a thorough and up-to-date review framework. The transition from the (rather meager) description of the issues related to the prevalence of problem gambling the introduction of the relationship of gambling with various constructs, and especially with psychopathologies and other health-related factors.
Some insights to report might be as follows:
https://doi.org/10.1155/2014/167438
https://doi.org/10.3389/fpsyt.2022.893861
it is necessary to repropose the definition of problem gambling according to the recent version of the Diagnostic and Statistical Manual of Mental Disorders
Line 46: The period is not well structured, please authors to explain it better.
Line 59 : it is necessary to repropose the definition of problem gambling according to the recent version of the Diagnostic and Statistical Manual of Mental Disorders
Line 83: The period is not well structured, please authors to explain it better.
Line 89: I beg the authors to explicate to the best of their ability the priming effect mentioned here
Line 111: check the space between: favord, it
4. State why the study with the sample of students from the Spanish is interesting. Does it present specificity or homogeneity compared to other nation? Are the results representative of an extended (national) condition or should they be considered specific to the student population of Valencia?
5. Review the graphics of the tables and images
6. Table 2 significant on correlations between gambling addiction and the other factors
7. adequately explain the regression model
8. Better explain in the text the relationship that emerged from the analyses conducted between relation of the various factors with gambling addiction
9. Rewrite the discussion with more critical considerations and reflections comparing the results that emerged in this study with others already conducted by other authors on the same topic.
10. The directions for future studies in the conclusion are unclear. These considerations should also rather be placed at the end of the discussion. It is recommended that the conclusion be reformulated and to provide for a detached section of the study limits.
Author Response
Comments and Suggestions for Authors
Dear Author,
first of all, I thank you for the opportunity to do the review of this interesting work. The objective of the study was to evaluate prevention and Harm Reduction Interventions of Gambling Behaviours.
The topic presented in the paper is interesting, as is the hypothesis tested by the model proposed by the authors
The technical description is timely and accurate, as is the graphical and tabular set-up. However, some points, if attended to by the authors with some additions, could further improve the quality of the work.
1) The introduction should be revised in translation, with a more synthesis, some passages are not smooth and there is too much redundancy and repetition.
Thank you very much for the reviewer's comment. We have revised the translation of the introduction, tried to synthesize the information and made some passages more fluid, tried to eliminate redundancy and repetition.
2) The description of the instruments is not accurate also needs, in my opinion, to be expanded.
We are grateful for the initial assessment of the manuscript, which has been reformulated by the pertinent observations of all the reviewers.
We have tried not to go too long in the description of the instruments to stick to the Healthcare standards. The objective was to present the measured variable, the type of items, the structure of the questionnaire and the reliability of the instrument for this sample.
3) The discussion should be revised in traslation
We appreciate the reviewer's comment. The translation has been peer-reviewed.
4)The conclusion to be expanded
Thank you very much for the reviewer's comments. We have expanded the conclusion.
5)The limitations of the study are lacking
We appreciate the reviewer's comments. New limitations have been included.
6) Abtract should be formatted and subdivided according to the journal rules.
We appreciate the reviewer's comments. The abstract has been adapted to the standards of the journal including Backgroun, Methods, Results and Conclusions.
Refer to and add the following articles:
https://doi.org/10.1155/2014/167438
https://doi.org/10.3389/fpsyt.2022.893861
We appreciate the reviewer's comments. The articles suggested by the reviewer have been referenced in the text.
Submission Date
16 July 2023
Date of this review
01 Aug 2023 17:14:22
Reviewer 3 New comments
It is necessary to repropose the definition of problem gambling according to
the recent version of the Diagnostic and Statistical Manual of Mental Disorders.
We appreciate the reviewer's comment. In the introduction section, it has been specified what is meant by problem gambling.
Line 46: The period is not well structured, please authors to explain it
better.
We appreciate the reviewer's comment. It has been specified since 2000. We think the reviewer is referring to line 59.
Line 59 : it is necessary to repropose the definition of problem gambling
according to the recent version of the Diagnostic and Statistical Manual of Mental
Disorders
What is meant by problem gambling has been included in the introduction. In the case of the study by Calado et al [8], it has been specified how gambling addiction is assessed and it has been specified that the study does not differentiate between problem and pathological gambling when discussing prevalence in different countries.
Line 83: The period is not well structured, please authors to explain it
better.
We appreciate the reviewer's comment. It has been specified since 2000. We think the reviewer is referring to line 65.
Line 89: I beg the authors to explicate to the best of their ability the
priming effect mentioned here.
We appreciate the reviewer's comment. The priming effect has been explained in more detail.
Line 111: check the space between: favord, it
We appreciate the reviewer's comments. It has been corrected.
- State why the study with the sample of students from the Spanish is
interesting. Does it present specificity or homogeneity compared to other nation? Are
the results representative of an extended (national) condition or should they be
considered specific to the student population of Valencia?
We thank the reviewer for his comments. In the limitations section, it is commented that the sample is specific and not representative, so the results cannot be generalized.
- Review the graphics of the tables and images
We are grateful for the reviewer's comments. The percentages in Table 1 have been corrected.
- Table 2 significant on correlations between gambling addiction and the other factors
We are grateful for the reviewer's comments. This has been corrected.
- adequately explain the regression model
We appreciate the reviewer's comment. We have included more information.
- Better explain in the text the relationship that emerged from the analyses
conducted between relation of the various factors with gambling addiction
Thank you very much for the reviewer's comment. This has been included in the discussion.
- Rewrite the discussion with more critical considerations and reflections
comparing the results that emerged in this study with others already conducted by
other authors on the same topic.
Thank you very much for the reviewer's comment. This has been included in the discussion.
- The directions for future studies in the conclusion are unclear. These
considerations should also rather be placed at the end of the discussion. It is
recommended that the conclusion be reformulated and to provide for a detached
section of the study limits.
We are grateful for the reviewer's comment. A limitations section has been included at the end of the discussion.
Round 2
Reviewer 2 Report
The path analysis should be tested by using a bootstrap method. Please consider to re-perform all the mediation analyses. In addition, the results of the analysis suggest a partial mediation. Please elaborate.
Author Response
Comments and Suggestions for Authors
The path analysis should be tested by using a bootstrap method. Please consider to re-perform all the mediation analyses. In addition, the results of the analysis suggest a partial mediation. Please elaborate.
Thank you very much for your interesting input. We have used a bootstrap method. We have rerun the analyses.
Finally, a simple mediation model with bootrstraping with 1000 samples, shown in Figure 1, was tested in which cognitive distortions acted as predictors of gambling addiction, with the maximum amount of money wagered being a mediator in this relationship. The model explained 34.5% of the variance (p < 0.01), and the indirect effect was 0.39 (p < 0.01). All the relationships shown were statistically significant (p<0.1), and the effect increased considerably when the maximum amount of money bet as a mediator was added.
Figure 1. Mediated regression model. **p < 0.1; c’ = direct effect; c = effect with mediator. Bootrstrap 1000 samples.
Reviewer 3 Report
The authors made all the necessary changes so that the article would be worthy of publication. I thank the authors and request that the article be published
Author Response
Comments and Suggestions for Authors
The authors made all the necessary changes so that the article would be worthy of publication. I thank the authors and request that the article be published.
Thank you very much for your comments and positive feedback.